# Role of N-Cadherin in Epithelial-to-Mesenchymal Transition and Chemosensitivity of Colon Carcinoma Cells

**DOI:** 10.3390/cancers14205146

**Published:** 2022-10-20

**Authors:** Veronika Skarkova, Barbora Vitovcova, Petra Matouskova, Monika Manethova, Petra Kazimirova, Adam Skarka, Veronika Brynychova, Pavel Soucek, Hana Vosmikova, Emil Rudolf

**Affiliations:** 1Department of Medical Biology and Genetics, Faculty of Medicine in Hradec Kralove, Charles University, Zborovska 2089, 500 03 Hradec Kralove, Czech Republic; 2Department of Biochemistry, Faculty of Pharmacy, Charles University, Akademika Heyrovskeho 1203, 500 05 Hradec Kralove, Czech Republic; 3The Fingerland Department of Pathology, Faculty of Medicine, University Hospital in Hradec Kralove, Charles University, Sokolska 581, 500 05 Hradec Kralove, Czech Republic; 4Department of Medical Biochemistry, Faculty of Medicine in Hradec Kralove, Charles University, Zborovska 2089, 500 03 Hradec Kralove, Czech Republic; 5Department of Chemistry, Faculty of Sciences, University of Hradec Kralove, Hradecká 1285, 500 03 Hradec Kralove, Czech Republic; 6Department of Toxicogenomics, National Institute of Public Health, Srobarova 49/48, 100 00 Prague, Czech Republic; 7Biomedical Center, Faculty of Medicine in Pilsen, Charles University, Alej Svobody 1655/76, 301 00 Pilsen, Czech Republic

**Keywords:** colon cancer, N-cadherin, primocultures, miRNA, EMT, chemosensitivity

## Abstract

**Simple Summary:**

The role of N-cadherin expression in epithelial-to-mesenchymal transition (EMT) and related aggressive tumor colon cancer cell phenotype was investigated using various in vitro and in vivo models. With the help of several standard laboratory techniques, it was verified that an artificially increased N-cadherin expression has only a limited reprogramming potential towards colon cancer cells unlike the case where colon cancer cells present with a naturally elevated presence of N-cadherin.

**Abstract:**

(1) Background: N-cadherin expression, epithelial-to-mesenchymal transition (EMT) and aggressive biological phenotype of tumor cells are linked although the underlying mechanisms are not entirely clear. (2) Methods: In this study, we used two different in vitro cell models with varying N-cadherin expression (stabilized lines and primocultures) and investigated their select biological features including the degree of their chemoresistance both in vitro as well as in vivo. (3) Results: We report that although enforced N-cadherin expression changes select morphological and behavioral characteristics of exposed cells, it fails to successfully reprogram cells to the aggressive, chemoresistant phenotype both in vitro as well as in vivo as verified by implantation of those cells into athymic mice. Conversely, primocultures of patient-colonic cells with naturally high levels of N-cadherin expression show fully aggressive and chemoresistant phenotype pertinent to EMT (in vitro and in vivo), with a potential to develop new mutations and in the presence of dysregulated regulatory pathways as represented by investigated miRNA profiles. (4) Conclusions: The presented results bring new facts concerning the functional axis of N-cadherin expression and related biological features of colon cancer cells and highlight colon cancer primocultures as a useful model for such studies.

## 1. Introduction

Colon carcinomas arise from colonic epithelial cells exposed to a series of genetic and epigenetic changes that in the context of local microenvironment drive their transformation into a fully malignant state. Similar to many other solid tumors, unfortunately, colon carcinomas are known to often locally invade surrounding tissues and subsequently spread to distant body sites via vascular and lymphatic metastases including the liver and the peritoneum among others [1]. Genotypic and phenotypic analyses of metastatic colon cancer cells have revealed their prevailing distinctness from the primary tumor cells and, namely, their high degree of biological aggressiveness associated with chemoresistance, often leading and contributing to premature death in a majority of patients [1].

Nowadays, we know that there are several identified reasons and particular mechanisms prompting the local and systemic spread of colon cancer cells and driving the changes in their behavior. The most relevant and central mechanism appears to be the peculiar transformation of these cells from non-motile, sedentary epithelial phenotype into mesenchymal-like form endowed with migratory, invasive and resistant features. This alteration occurs throughout a genetic reprogramming process known as epithelial-to-mesenchymal transition (EMT). EMT has several physiological roles in our organism, i.e., it occurs in embryogenesis, wound healing and it is also involved in tissue repair [2,3]; however, it may underlie and lead to pathological consequences too (organ fibrosis and/or cancer) [4,5,6]. Accordingly, context-dependent EMT can be stimulated by various signals that in the case of cancer-related EMT further depend on target malignant cells. Thus, in colon carcinoma cells, EMT is known to be induced by platelet-derived growth factor (PDGF) through the nuclear translocation of β-catenin in a WNT-independent manner [7]. A similar process may also be at work during epidermal growth factor (EGF) and tumor growth factor beta (TGFβ)-induced EMT [8]. Besides these known external stimuli, some intracellular signals appear to possess similar potential too as for instance protein tyrosine phosphatase PRL3 has been reported to activate phosphatidylinositol 3-kinase/protein kinase B (PI3K/AKT), thereby increasing the degradation of phosphatase and tensin homolog (PTEN) and activating Snail1 leading to EMT [9].

Although of different origin and context, the chief events of the EMT process are very similar. Thus, on the one hand concerned cells lose their typical epithelial cell polarity and cell/cell contacts while downregulating expression of the specific cytoskeletal and junctional proteins (cytokeratins, claudins, occludins, E-cadherin). On the other hand, cells increase the expression of mesenchymal adhesive and cytoskeletal proteins fibronectin, N-cadherin, and vimentin, change their cytoskeletal architecture and upregulate matrix metalloproteases enabling cell migration and invasion along with elevated resistance to cell death [10,11].

Biological aggressiveness and chemoresistance of cells undergoing EMT have been described in several types of tumors including colon cancer [12,13,14,15,16]. In addition, chemotherapy-induced EMT in cancer cells may lead to developed chemoresistance as demonstrated in colon cancer too [17]. Accordingly, changes in the expression of chief EMT intracellular markers E-cadherin and in particular N-cadherin have been associated with important clinicopathological features and survival of patients. To this extent, the loss of E-cadherin is associated in humans with worse clinical prognosis whereas higher expression of N-cadherin is directly related to tumor aggressiveness, its systemic spread and resistance to cell death [18,19,20,21].

The process of EMT underlies an interplay of complex molecular pathways that involve among other players microRNAs (miRNAs such as the miR-200 or miR-21 families), epigenetic and posttranslational regulators along with alternative splicing events [22]. However, the key hallmark in these processes seems to be the shift from E- to N-cadherin expression [23]. Given the emerging role of N-cadherin in colon cancer development and its clinicopathological course, it represents a promising target of not only diagnosis but also intervention. Here, data from prostate cancer indicate that peptides and mouse monoclonal antibodies targeting N-cadherin proved some efficacy in inhibiting prostate cancer metastasis and enhancing cancer cell sensitivity to chemotherapeutic agents [24].

Therefore, in the present work we focused on the role of N-cadherin in modulation of selected colon cancer cell biological characteristics and behavior both in vitro and in vivo. We used and compared two different in vitro models with varying N-cadherin expression, i.e., colon cancer cell lines HCT8 and SW480 with artificially upregulated expression of N-cadherin, and colon cancer cells with differing N-cadherin levels obtained from patients. Moreover, we attempted to investigate the relationship between mesenchymal phenotype of cells (here represented by an increased expression of N-cadherin) and the degree of their chemoresistance.

## 2. Materials and Methods

### 2.1. Cell Lines

Human epithelial adenocarcinoma colon cancer cell lines SW480 and HCT8 were purchased from ATCC (LGC Standards, Poland). Frozen stock cells were thawed for every set of experiments (duration 3–9 weeks) and cultured in DMEM supplemented with 10% FBS and 0.5% penicillin/streptomycin. Cells were cultured in incubators with a humidified atmosphere containing 5% CO_2_ at 37 °C. The absence of mycoplasma contamination was periodically checked in all cell lines used.

### 2.2. Clinical Samples

Cell primocultures labeled as 36B, 39B, 42B, 43B, 44B, 46B, 47B, 48B and 54B were derived from tissue samples obtained from patients who underwent surgery for colorectal carcinoma at University Hospital in Hradec Kralove. The study was approved by local ethics committee (Reference No. 201604S O3P—attached as a Appendix A) and patients gave their written consent (attached as a Appendix A). All samples included in the study differed in tumor stage and grade. The amount and quality of sampled materials used for cell culture derivation varied based on individual surgical specimens. The primoculture derivation procedure and further manipulation with cell cultures was described before [25].

### 2.3. N-Cadherin Plasmid Preparation and Cell Transfection

Total RNA was isolated from human healthy liver tissue from our sample collection using Tri Reagent (Molecular Research Centre). cDNA was synthesized using the ProtoScript II reverse transcriptase and oligo (dT). The full-length coding sequence of N-cadherin covering the complete coding region of the *Homo sapiens* gene was obtained from the NCBI database (accession No. NM_001792.4:425-3145). Primers used for amplification were as follows: F: TAACTCGAGATGTGCCGGATAGC and R: TCATCTAGATCAGTCATCACCTCCA; these include restriction sites for XhoI and XbaI, respectively, for direct cloning into pCI Mammalian expression vector (Promega). The encoding region was amplified using Phusion polymerase (Thermo Fisher Scientific), checked by agarose electrophoresis, purified by Zymo Research PCR purification Kit, digested by XhoI and XbaI and ligated into pCI vector using T4 DNA ligase (Promega) to yield Ncadh-pCI transfection vector. The sequence of the whole insert was verified by Sanger sequencing and primer walking by Eurofins genomics (Germany).

### 2.4. WST-1 Proliferation Assay

The effect of chemotherapeutics irinotecan (IT) and oxaliplatin (OPT) on viability of colon cancer cell lines HCT8, SW480 and their N-cadherin plasmid treated variants, as well as on derived colon cancer primocultures 36B, 39B, 42B, 43B, 44B, 46B, 47B, 48B and 54B was evaluated by WST-1 assay. This colorimetric assay is based on the cleavage of tetrazolium salt to colored formazan by mitochondrial dehydrogenases in viable cells. The viability of tested cells is quantified by measuring activity of mitochondrial enzymes. Colon cancer cells were exposed to various concentrations of IT and OPT (all diluted in cultivation medium) for 48 h in 96-well plates. At the end of each tested interval, cells were rinsed with PBS and WST-1 solution (diluted according to manufacturer’s recommendations) was added to each well for further 2 h. The absorbance was measured at 450 nm with 650 nm of reference wavelength by Tecan Infinite M200 spectrophotometer (Tecan, Switzerland).

### 2.5. xCELLigence Proliferation Assay

Proliferation of all employed colon cancer cells was evaluated using xCELLigence Real-Time Cell Analyzer (RTCA). Electrical impedance measured by microelectrodes located in the bottom of the E-plate provides label-free dynamic monitoring of cell proliferation in real time and its relative changes are displayed by arbitrary units called the “cell index”. First, 90 µL of cultivation medium added into each well in the E-plate served for background measurement. Further, 100 µL of cell suspension (25,000 cells/mL) was added into each well in quadruplicate. The attachment, spreading and proliferation of the cells were monitored every hour for 99 h of cultivation. The data are expressed as % of untreated control.

### 2.6. xCELLigence Migration Assay

Cell migration of all employed colon cancer cells was measured using xCELLigence Real-Time Cell Analyzer (RTCA) in CIM plate 16, which serves as a modified Boyden chamber with upper and lower compartment. Serum-free cultivation medium with 1% BSA was pipetted into upper chambers of the plate and medium supplemented with 10% FBS was added into lower chambers. Both chambers were next locked together to form a tight seal and left for 1 h before background measurement. Thereafter, cell suspension (300,000 cells/mL) in serum-free medium with 1% BSA was added into the upper chamber compartment. Cell migration was recorded every 10 min for 24 h as a “cell index”. The data in graphs are expressed as % of untreated control.

### 2.7. Fluorescence Microscopy

SW480 and HCT8 colon cancer cells were treated with N-cadherin plasmid in cytospine chambers, then fixed with 2% paraformaldehyde (20 min, 25 °C), rinsed with PBS permeabilized, and blocked with 1% Triton X and 5% BSA in PBS (30 min, room temperature). Cells were incubated with a primary antibody against N-cadherin (D4R1H XP^®^ Rabbit mAb, Cell Signaling) at 4 °C overnight. Then, cells were washed three times with cold PBS (5 min, 25 °C) and were incubated for additional 1 h (room temperature) with Alexa Fluor 488-labeled anti-rabbit antibody. Thereafter, cells were rinsed three times with PBS and post-labeled with DAPI (10 µg/mL). The specimens were mounted into the Prolong Gold mounting medium (Invitrogen-Molecular Probes, Inc., Carlsbad, California, CA, USA) and examined under a fluorescent microscope (Nikon Eclipse E 400 (Nikon Corporation, Kanagawa, Japan)). Captured images were analyzed using LUCIA DI Image Analysis System LIM 4.2 (Laboratory Imaging Ltd., Prague, Czech Republic). All the samples were tested in duplicates in three independent experiments.

### 2.8. Phase Contrast Microscopy

Morphology and architecture of colon cancer cells was recorded in standard 6-well plate under Olympus IX-70 phase contrast microscope using various magnifications and at different time intervals.

### 2.9. Western Blot Analysis

HCT8, SW480 cell lines and their N-cadherin plasmid treated variants as well as derived colon cancer primocultures 47B and 48B were grown in 6-well plates (150,000 cells/mL). Then, the cells were washed with PBS and harvested in ice-cold lysis buffer. Tumors excised from mice (both variants of HCT8 tumors and 47B, 48B tumors) were homogenized using Tissue lyzer (2 cycles—25 vibration/s; 4 °C; 1 min; Qiagen, Germantown, MD, USA) in ice-cold lysis buffer. Protein content of resulting cell lysates was determined by BCA assay. Detailed description of SDS-PAGE of diluted cell lysates (30 µg/well), protein transfer to PVDF membrane, antibody incubation and protein detection is summarized in [26]. The following primary antibodies from Cell signaling technology were used: polyclonal rabbit anti-E-cadherin 1:2000; polyclonal rabbit anti-N-cadherin 1:1500; polyclonal rabbit anti-vimentin, 1: 2000; monoclonal rabbit anti-CD44, 1:2000. Monoclonal rabbit anti-MRP1, 1:2500; monoclonal rabbit anti-MDR1, 1:2500, monoclonal rabbit anti-FAK, 1:5000; monoclonal rabbit anti-GEF-H1, 1:2000 and housekeeping monoclonal mouse α-tubulin, 1:10,000. Housekeeping monoclonal mouse β-actin, 1:10,000 was from Sigma Aldrich. Samples were analyzed by Azure Biosystems c600 and semi-quantitative analysis was performed using program Azure spot.

### 2.10. RNA Extraction, cDNA Synthesis, Primer Design, Quantitative Real-Time RT-PCR

Total RNA was isolated from HCT8 and SW480 cell lines and their N-cadherin plasmid treated variants as well as from derived colon cancer primocultures using Direct-zol RNA MiniPrep kit according to the manufacturer’s instructions (ZymoResearch, Irvine, CA, USA). Tumors excised from mice (both variants of HCT8-originating tumors and 47B, 48B-originating tumors) were homogenized using Tissue lyzer (2 cycles—25 vibration/s; 4 °C; 1 min; Qiagen, Germantown, MD, USA) in TriReagent. RNA concentration and its purity were determined using NanoDrop 2000 (Thermo Fisher Scientific). All samples had an absorption ratio A260/A280 greater than 1.8. RNA integrity number (RIN) was determined using Agilent 2100 Bioanalyzer and cell line samples with RIN higher than 9.0, resp. tissue samples with RIN higher than 8.0 were used for further analysis. First strand cDNA synthesis and qPCR analysis were performed in LightCycler^®^ 96 Instrument (Roche Life Science). Primers were designed manually, and their sequences are provided in a Appendix A. Calculations were based on delta-delta Cq method (13) and expressed as fold change of the treated groups relative to the control. Beta-2-microglobulin (B2M) was used as a reference gene for mRNA analysis and miR-16 was used as a reference gene for miRNA analysis.

### 2.11. MiRNA Isolation from Plasma

Total RNA, including the miRNA fraction, was isolated from 300 µL of plasma samples by miRNeasy Serum/Plasma Kit (Qiagen) according to manufacturers’ protocol with the following modifications: (1) the volumes of Qiazol and chloroform were adapted to a larger volume of source material, (2) 20 µg of glycogen (ThermoFisher, Pardubice, CZ. USA) was added to each sample before loading them onto silica-membrane, (3) the efficiency of isolation was monitored by spike-in control cel-miR-39-3p (20 fmol per isolation, ID: MC10956, ThermoFisher). RNA concentration was determined with Quant-iT RiboGreen RNA Quantitation Assay Kit (ThermoFisher, Pardubice, CZ, USA) with the low-range protocol using Infinite M200 multiplate reader (Tecan Group Ltd., Männedorf, Switzerland). Complementary DNA was generated with TaqMan MicroRNA Reverse Transcription Kit (ThermoFisher, Pardubice, CZ, USA) by mixing the extracted RNA (1 μL, 10× diluted) with 5× cel-39 primers (1.5 μL), 10× Reverse Transcription Buffer (0.75 μL), 10 mM dNTP (0.75 μL), RNase Inhibitor 20 U/μL (0.095 μL), MultiScribe Reverse Transcriptase 50 U/μL (0.5 μL) and nuclease-free water to final volume 7.5 μL. This mixture was incubated in thermal cycler at 16 °C for 30 min, at 42 °C for 30 min, at 85 °C for 5 min and then at 4 °C. The subsequent control qPCR was carried out using 2× TaqMan Universal Mastermix II (2.5 µL), 20× TaqMan Small RNA Assay for spike-in control (ID: 000200) (0.25 µL), nuclease-free water (0.25 µL), and 6× diluted cDNA with the following cycling program: initial denaturation at 95 °C 10 min, 40 cycles of 95 °C for 15 s and 60 °C for 60 s in ViiA7 Real-Time PCR System (ThermoFisher, Pardubice, CZ, USA).

### 2.12. miRNA Microarray Profiling

Expression levels of miRNAs isolated from plasma and tissue samples were assessed by SurePrint G3 Human miRNA Microarrays (8 × 60 k, Release 21.0, Design ID 070156; Agilent Technologies, Santa Clara, CA, USA) according to manufacturers’ protocol. Plasma levels of miRNA are generally low and thus the volume of the sampled plasma was used to standardize sample amount used for subsequent analyses. Briefly, 3.1 µL of total RNA isolated from plasma samples or 2 µL of total RNA (50 ng/µL) isolated from tumor samples was labeled with Cy3 using the miRNA Complete Labeling and Hybridization Kit and spiked with synthetic control miRNAs for accessing labeling performance (Agilent Technologies). The Cy3-labeled samples were hybridized for 20 h at 55 °C in a rotator oven. After washing steps, the array slides were scanned using AgilentSureScan microarray scanner (Agilent Technologies). Expression data and Quality Control (QC) Reports were extracted from the scanned images using Feature Extraction version software (version 12.1, Agilent Technologies). The data generated in this study are publicly available in Gene Expression Omnibus (GEO) at GSE205027.

### 2.13. LC-MS Analysis

HCT8, SW480 cell lines and their N-cadherin plasmid treated variants as well as derived colon cancer primocultures 47B and 48B were plated in 6-well plates (150,000 cells/mL). They were exposed for 4, 8 and 24 h (HCT8 and SW480), resp., for 4 and 8 h (47B and 48B) to IT and OPT concentrations corresponding to their determined IC_50_ values. The concentration of DMSO in medium was 0.1%. Cells were collected into 500 µL of sterile distilled water. The total protein amount in the cell suspension homogenates was determined with a bicinchoninic acid (BCA) assay (Sigma Aldrich) according to the manufacturer’s protocol.

Detection of IT and its metabolite SN-38 was performed on the Agilent 1290 Infinity II UHPLC system coupled to the Agilent 6470 QqQ mass spectrometer. Chromatographic conditions were maintained at gradient elution of 0.4 mL/min by 0.1% formic acid in water and methanol (0–0.1 90:10, 0.1–3.0 gradient to 70:30, 3.0–4.0 10:90, 4.0–5.0 90:10), thermostated autosampler set to 15 °C and column thermostat equipped with the Zorbax Eclipse plus RRHD C18 2.1 × 50 mm, 1.8 µm (PN 959757-902) column kept to 30 °C. MS source parameters were set to the following: drying gas 260 °C at 8 L/min, sheath gas 400 °C at 12 L/min, nebulizer pressure 25 psi, capillary voltage 4000 V and nozzle voltage 300 V. Transitions of [M + H]+ ions m/z were detected with setting of dwell time 20 ms, cell acceleration 4 V, fragmentor 172 V for 587→195, 167, 124 and 110 (collision energy—CE 32, 48, 40 and 40 V).

Detection of OPT was performed on the Agilent 1290 Infinity II UHPLC system coupled to the Agilent 6470 QqQ mass spectrometer. Chromatographic conditions were maintained at gradient elution of 0.4 mL/min by 0.1% formic acid in water and methanol (0–0.5 95:5, 0.5–3.0 gradient to 5:95, 3.0–4.0 5:95, 4.0–5.0 95:5), thermostated autosampler set to 15 °C and column thermostat equipped with the Zorbax Eclipse plus RRHD C18 2.1 × 50 mm, 1.8 µm (PN 959757-902) column kept to 30 °C. MS source parameters were set to the following: drying gas 200 °C at 6 L/min, sheath gas 400 °C at 10 L/min, nebulizer pressure 35 psi, capillary voltage 3000 V, nozzle voltage 300 V. Transitions of [M + H]+ ions m/z were detected with setting of dwell time 160 ms, cell acceleration 4 V and fragmentor 105 V for 398→306, 96 and 79 (collision energy—CE 24, 40 and 60 V).

### 2.14. Preparation of Tumor-Bearing Mice Model

Female Foxn1-nu athymic immunodeficient mice weighing 27–30 g were purchased from Anlab, Czech Republic. They were given a standard sterilized diet and water ad libitum. The cell suspensions used for transplantation were the same as described above for in vitro assays. For the cancer cell implantation, the cell suspensions were diluted with PBS to the final concentration 3 × 10^6^ cells for HCT8 cells and its HCT8_N-cad variant as well as colon cancer primocultures 47B and 48B.

HCT8 and HCT8_N-cad variant cancer cells were implanted s.c. on the right and left side of the back in five mice. The size of the tumors and health condition were periodically checked and on 21st day, the mice were anesthetized with isoflurane. 47B and 48B cells were implanted s.c. on the right and left side of the back in three mice. The size of the tumors and health condition were periodically checked and on 40th day, the mice were anesthetized with isoflurane. Blood samples (1 mL from each mouse) were collected by posterior vena cava puncture using K3EDTA blood collection tubes for miRNA analysis (Sarstedt, Brand-Erbisdorf, Germany). The collected blood samples were centrifuged twice for 5 min at 300× *g* for plasma separation and stored in −80 °C until analysis. Tumors were excised from mice, weighed and stored until further analysis (in formalin at room temperature for IHC analysis, in Trisol at −80 °C for RT-PCR analysis and in lysis buffer at −20 °C for Western blot analysis).

### 2.15. Immunohistochemical Analysis

All surgical samples were fixed in formalin and made into paraffin blocks using standard paraffin embedding procedure. Four µm thick sections were cut from paraffin blocks, mounted on positively-charged slides. The primary antibodies used for immunohistochemical detection were E-cadherin (Ventana, Dako, clone NCH-38, predilute 1:50), N-cadherin (Ventana, Leica, clone IAR06, predilute 1:100), vimentin (Ventana, Dako, clone V9, predilute 1:400) and Ki-67 (Ventana, ready to use). Antigen retrieval was performed in a water bath for 20 min at 97 °C at pH 6 (buffer S1700, Dako, Glostrup, Denmark). Endogenous peroxidase activity was inhibited by immersing the sections in 3% hydrogen peroxide. After incubation with the antibody, the sections were subjected to EnVisionTM FLEX (Dako, Glostrup, Denmark). Finally, slides were counterstained with haematoxylin, mounted and examined.

### 2.16. Mutational Analysis

The DNA was extracted from paraffin-embedded tissue blocks by the Cobas DNA Sample Preparation Kit (Roche) according to manufacturer’s protocol. Mutation analysis was performed by multiparallel sequencing (NGS). Indexed Illumina NGS library was constructed from 100 ng tumor DNA by KAPA Library Preparation Hyper Plus Kit (Kapa Biosystems). Hybrid selection was performed with a custom SeqCap EZ Choice Library (Roche NimbleGen). The library was designed using genome build hg19 NCBI. Build 37.1/GRCh37 input genomic regions are listed as follows: AKT, BRAF (exons 11,15), BRCA1, BRCA2, EGFR (exons 18, 19, 20, 21), ESR, GNAS (exon 8), KRAS, KIT, MLH1, MSH2, MSH6, NRAS, PDGFRA (exons 8, 10, 12, 14, 18), PIK3CA, PMS2, PTEN, RET and TP53.

Paired-end cluster generation and sequencing were performed according to standard protocols from Illumina, using v2 kits. Sequencing data analysis and variant classification were performed by NextGENe software (Softgenetics) with minimum 5% variant allele frequency filtering.

### 2.17. Statistical Analysis

Data from all analyses and assays are expressed as an average ± SD from at least two experiments. The concentration of chemotherapeutic IT and OPT causing a 50% decrease of cell viability (IC50 value) was determined by Graph Pad Prism 7.0. Statistical analyses of the data from immunoblotting, RT-PCR, proliferation assay, migration assay, WST-1 assay and LC-MS analysis were carried out using TWO-WAY analysis of variance (ANOVA) followed by Sidak’s multiple comparison test significant at *p* ˂ 0.05 using GraphPad Prism 7.0.

The microarray data analysis was performed by the Agilent GeneSpring GX software (version 14.9, Agilent Technologies). Files with expression data from tumor and plasma samples were loaded and analyzed separately. Data were normalized using the quantile normalization, without baseline transformation, and filtered: (1) on flags, i.e., only miRNAs with positive expression (“detected”) in three of the eight samples (HCT8 human colon cancer cell line samples) or in two of four samples (colon cancer primocultures) were included in further analysis; (2) on expression levels (20–100th percentile) under the same conditions; and (3) on error, i.e., only miRNAs with coefficient of variation among replicates < 50% were included.

Differences in the miRNA expression profile between HCT8 cells (HCT8 CTRL) and HCT8 cells with overexpressed N-cadherin (HCT8_N-cad) grown in mice, between plasma samples obtained from the same mice, and between colon cancer primocultures 47B and 48B were evaluated using a moderated t-test and the fold-change analysis. The Benjamini–Hochberg false discovery rate (FRD) test was used for the multiple testing correction. Significance was defined by the absolute fold change ≥2 and a corrected *p* value (adj. *p* value) less than 0.05.

Gene Ontology (GO) enrichment analysis and Kyoto Encyclopedia of Genes and Genomes (KEGG) pathway analysis of differentially expressed miRNAs were carried out with the miRNet v2.0 [27] using experimentally validated gene targets obtained from TarBase v8 [28] and miRTarBase v8 [29]. The significance threshold for GO terms in biological processes (BP) and KEGG pathways was defined by FDR-corrected *p* values (adj. *p* < 0.05).

All microarray data was MIAME (Minimum Information About a Microarray Experiment) compliant [30] and has been submitted to the Gene Expression Omnibus (GEO, platform ID: xxx)

## 3. Results

### 3.1. Isolation and Biological Characteristics of Stabilized Colon Cancer Cell Line-Derived Colon Cancer Primocultures

Two selected and employed colon cancer cell lines SW480 and HCT8 were successfully transfected with N-cadherin plasmid (Figure 1 and Figure 2 for Appendix A). These original and N-cadherin-transfected cells represented our first model for subsequent evaluation of individual colon cancer cell characteristics.

Nine different primocultures isolated from patient samples of colon cancer were established via a standard protocol and their N-cadherin expression as well as general sensitivity to commonly used chemotherapeutics irinotecan (IT) and oxaliplatin (OPT) were investigated. It was found that individual primocultures expressed N-cadherin to a varying degree and differed in their response to employed cytostatic compounds too.

The most significant differences in the mentioned parameters were found in two primocultures labeled 47B and 48B, respectively (Figure 3 for Appendix A). Sample cells 47B exhibited a high N-cadherin protein level whereas in 48B sample cells no N-cadherin protein was detected. Similarly, 48B cells were tested as more sensitive to IT and OPT as compared to the 47B cells. Cells of both 47B and 48B primocultures originated from excised tumors of the same localization and the same grade and both harbored a mutation in the *K-RAS* gene (c.35G > T—in 47B and c.38G > A in 48B). Still, some clinicopathological features of patients identified with the examined tumors differed. The patient 47B was positive for lymphangioinvasion, vascular invasion and perineural spread of tumor unlike the patient 48B. Given these differences corresponding to the N-cadherin status in the first model cells, the primocultures 47B and 48B were used as second models in further experiments.

First, proliferation and migration of all tested cell cultures (original and transfected variants of HCT8 and SW480, and the derived primocultures 47B and 48B) were measured. No differences in proliferation and migration in both the original and transfected variant were found in HCT8 cells (Figure 4 for Appendix A). In the SW480 cell line, the transfected variant (SW480_N-cad) cells showed a significantly increased proliferation at a later treatment period as compared to the original ones (Appendix A). SW480 cells do not migrate and N-cadherin transfection did not change it. Proliferation of 47B and 48B cells markedly differed during the first 50 h, with 47B cells showing a higher rate of growth and division contrary to 48B cells. Later, this trend ended and both types of cells continued in their proliferation with the same rate as evidenced by their nearly identical growth curve (Appendix A). Migration activity of 47B cells was significantly higher than in 48B cells for almost the entire investigated time period (Appendix A).

Selected markers of EMT and cell migration potential in both variants of the tested cells and derived primocultures were evaluated using RT-PCR (Figure 1A–C) and Western blot analysis (Figure 1D). As expected, significant expression/presence of N-cadherin was confirmed in both transfected HCT8, SW480 cell lines and in 47B cells on mRNA and protein levels. The HCT8_N-cad variant exhibited a reduced expression of transcription factors ZEB1 and ZEB2 while E-cadherin and miR-200a, b and c were markedly upregulated as compared to the original HCT8 cells (Figure 1A). In contrast, in the SW480_N-cad variant cells no significant differences in the expression of ZEBs, E-cadherin and miR200a, b or c were found although their levels were to various degrees reduced (Figure 1B). Expressions of ZEBs and miR200a were significantly increased in 48B cells. MiR200b and miR200c were not detected in these samples (Figure 1C). Measured protein abundances of examined markers in model cells positive for higher N-cadherin expression (HCT8_N-cad variant, SW480_N-cad variant and 47B cells) revealed an increased presence of GEF-H1 in the HCT8_N-cad variant, FAK in the SW480_N-cad variant and CD44 in 47B cells (Figure 1D). On the other hand, E-cadherin and CD44 levels were markedly decreased in the SW480_N-cad variant. Moreover, vimentin presence was significantly enhanced in 48B cells while being suppressed in N-cadherin-positive 47B cells (Figure 1D).

### 3.2. Chemosensitivity of Stabilized Cell Lines and Derived Colon Cancer Primocultures

Next, all the tested cell cultures of both models were treated with several concentrations of IT and OPT for 48 h. Obtained data clearly show that HCT8 and SW480 cell lines responded to the employed compounds in an entirely different way than cells derived from patient tumor samples. N-cad variants of both SW480 and HCT8 cells were more sensitive to IT and OPT than their original counterparts. In contrast, primoculture 47B with naturally higher N-cadherin expression was 21.4 times less sensitive to IT and 8.7 times less sensitive to OPT, in comparison with the primoculture 48B having no detectable N-cadherin levels. The summarized results of chemosensitivity of the tested colon cancer cell cultures to IT and OPT are shown in Table 1.

In order to elucidate the discovered differences in chemosensitivity of tested cell cultures, attention was next paid to the intracellular metabolism of IT and OPT using LC-MS analysis of parent compounds as well as IT metabolites SN-38 and 7-ethyl-10-[4-(1-piperidino)-1-amino] carbonyloxycamptothecin (NPC). As shown in Figure 2, changes in drug levels were detected at 4 h, 8 h and 24 h after treatment. While the levels of IT were stable during all the tested time intervals in both variants of HCT8 and SW480 cell lines, concentrations of its metabolites significantly changed in a time- and cell type-dependent manner. The measured levels of SN-38 and NPC were significantly decreased in HCT8_N-cad cells in comparison with HCT8 cells (Figure 2A). In SW480_N-cad cells high levels of SN-38 metabolite were detected at 24h of treatment, 1.9 times more than in original cells. On the other hand, NPC metabolite levels were not present or time-dependently decreased in both SW480 subclones/sublines (Figure 2C).

Due to the lower number of cells of 47B and 48B primocultures available for LC-MS analysis, they were limited to time intervals of 8 h and 24 h only. Intracellular concentrations of IT and SN-38 and NPC metabolites were significantly increased at both time intervals in 48B cells as compared to 47B culture (Figure 2E).

The intracellular levels of OPT increased time-dependently in both HCT8 cell variants and reached the maximum at 24 h after treatment (Figure 2B). A similar trend occurred in the SW480 cell line; however, SW480_N-cad cells accumulated 1.7 times higher OPT concentration at the 24 h interval (Figure 2D). Whereas the intracellular concentration of OPT increased very slightly at the 24 h interval in 47B cells, OPT levels in 48B cells showed at 8 h of treatment 3 fold increase and 2.5 fold increase at 24 h treatment (Figure 2F).

### 3.3. Behavior and Phenotypical Analysis of Cells Obtained from Colonic Tumors In Vivo

The data obtained from in vitro experiments were next verified and compared to an in vivo study. HCT8 cells, resp. HCT8_N-cad cells were s.c. implanted into both sides of athymic immunodeficient Foxn1-nu mice. Twice a week, the health status of mice and growth of tumors were monitored. Tumor samples were excised on the 21st day and the size and weight of individual tumors were determined. Implanted parent HCT8 cells (lower N-cadherin levels) produced smaller size tumors as compared to HCT8_N-cad cells (Figure 3B). Implantation of 47B and 48B primoculture cells into the athymic immunodeficient Foxn1-nu mice took place too. The growth of the tumors was terminated on the 40th day. Implanted 48B cells (no detectable N-cadherin levels) produced significantly larger tumors than implanted 47B cells (higher level of N-cadherin) (Figure 3C).

Cells of excised tumors were next analyzed for the expression of selected EMT, migration, invasion and chemosensitivity markers using RT-PCR and Western blot analysis and obtained data are summarized in Figure 4. In the case of tumors originating from implantation of HCT8 cells, the expression of most examined markers was not changed compared to the original cell line variant (HCT8/HCT8_N-cad). The only difference confirmed on mRNA and protein levels concerned N-cadherin as well as an increased vimentin expression (mRNA level). Tumors originating from implantation of 48B cells still showed low levels of N-cadherin (mRNA and protein) as compared to 47B-originating tumors. In addition, significantly decreased CD44, MMP2 or MMP9 protein levels were also found in 48B-originating tumors, similar to the original 48B cells (Figure 4A,B). On the other hand, elevated levels of drug resistance markers MDR1 and MRP1 and motility markers FAK and GEF-H1 were detected in 48B tumors in comparison with 47B tumors (Figure 4C).

Excised tumors were next analyzed immunohistochemically to determine the expression of E-cadherin, vimentin and proliferative activity of cells. While the expressions of both analyzed markers as well as the proliferation in cells originating from HCT8 tumors were almost similar, in 47B- and 48B-originating tumors significant differences were observed. Proliferation activity as measured by Ki-67 marker was higher in 48B tumors (Figure 5). Expression of epithelial marker E-cadherin was elevated in 48B tumors, while mesenchymal marker vimentin was significantly expressed in 47B tumors, which confirmed the results from previous analyses.

### 3.4. Mutational Analysis of Cells from Colon Cancer Tumors In Vivo

Excised tumor samples were further analysed for the presence of mutations. While mutation analysis of HCT8 and HCT8_N-cad tumors did not reveal any differences or new mutations, 47B- and 48B-originating tumors showed mutation diversity. Cells from both 47B- and 48B-derived tumors harbored mutations in *APC*, *KRAS* and *TP53* genes although with distinct variant types. In addition, in 48B-originating tumor cells a unique mutation was found in *PIK3CA*, and *RET* genes (Table 2).

### 3.5. miRNA Analysis of Cells from Colon Cancer Tumors In Vivo

The miRNA arrays were performed with RNA isolated from plasma and tumor samples of mice with implanted HCT8 cells with N-cadherin overexpression (HCT8_N-cad) and HCT8 control cells (HCT8 CTRL) with four biological replicates in each compared group. We found 22 differentially expressed miRNAs in HCT8_N-cad tumor samples (16 upregulated and 6 downregulated, |FC| ≥ 2) in comparison to HCT8 CTRL tumor samples, with only hsa-miR-3620-5p being significantly downregulated (FC −28.52, *p* < 0.001, Appendix A).

Four miRNAs were found to be differentially expressed in HCT8 NC plasma samples in comparison to CTRL samples (one upregulated, three downregulated, |FC| ≥ 2), with only hsa-miR-3680-3p being significantly downregulated (FC -20.57, *p* < 0.001, Appendix A). Experimentally validated targets of hsa-miR-3620-5p and hsa-miR-3680-3p according to miRTarBase v8 and TarBase v8 databases are included in the Appendix A.

Next, the differences in miRNA expression profiles of mice-implanted cells of two colon cancer primocultures 47B and 48B were analyzed (47B tumors and 48B tumors both in two replicates). In total, 109 miRNAs were differentially expressed (61 upregulated and 48 downregulated) in 47B tumor cells in comparison to the 48B sample (Figure 6a,b and Appendix A). For all 109 miRNAs, we found 13162 experimentally validated targets. After filtering step by the term “Luciferase assay” as validation method, we identified 824 validated targets for 54 of 109 miRNAs. By the GO and KEGG analysis of these 824 genes, we identified more than 96 significant biological process terms and 84 significant pathways (adj. *p* < 0.05) (Figure 6c, Appendix A).

## 4. Discussion

N-cadherin belongs to the calcium-dependent adhesion molecule family of cadherins involved in morphogenetic processes occurring primarily during development. Moreover, in adulthood, N-cadherin is required for synaptic functions, vascular stability and bone homeostasis, which it influences via its expression in bone, neural or endothelial cells [29,30]. In epithelial cells, N-cadherin is absent or present at best at low levels and its increased and/or aberrant expression have been associated with malignant transformation in many tissues including the ones of the digestive tract [20,31,32]. Elevated levels of N-cadherin in solid tumor cells are nowadays recognized to be related to select clinicopathological features and independently associated with poor patient prognosis [19,20,21]. In the cellular context, increased N-cadherin expression is known to stimulate the migratory and invasive potentials of concerned cells for instance via fibroblast growth factor receptor 1 (FGFR-1) or canonical Wingless and Int-1 (WNT) signaling circuits [33,34]. Its robust presence in the cell is also associated with developed chemoresistance with poorly understood mechanisms, although activation of prosurvival protein Bcl-2, protein kinase B or decoy receptor-2, modulation of Sonic Hedgehog signaling and generally acquired stem cell phenotype have been proposed to this extent [35,36,37,38].

In our work, we sought to compare biological characteristics of colon cancer cells with high (enforced or naturally present) and low expression of N-cadherin both upon in vitro as well as in vivo conditions. Our experimental results show that upon enforced expression of N-cadherin, original epithelial morphology, the proliferative and migratory activities of treated cells (SW480 and HCT8) positively change although these changes (in particular cell growth and migration) are not consistent and appear to be time-limited and cell-type dependent. In addition, this inconsistency is also reflected in our observed expression spectrum of investigated EMT markers. This may well signify that such N-cadherin expression, although considered as a critical step in EMT and its driven biological aggressiveness [39], may not insure the successful completion of this process in target cells. Such assumption is further supported by our paradoxical observation concerning sensitivity of thus manipulated colon cells to model chemotherapeutics of IT and OPT that could not be possibly explained by differences in intracellular metabolism of these compounds. These findings were further confirmed in vivo, where most compared parameters including cell mutational profiles, proliferative activities and expressions of select EMT markers in tumors originating from cells with increased N-cadherin expression versus those from control cells did not reveal any significant differences. The only found distinctions concerned two significantly downregulated miRNAs—intracellular hsa-miR-3620-5p and plasma hsa-miR-368-3p. Both mentioned miRNAs have been reported to occur in an upregulated or downregulated state in some malignancies [40] but their relation to N-cadherin expression, EMT and associated biological features of malignant cells remain unknown. In this light, their downregulation in the present model may simply reflect an overall changed regulatory environment of manipulated cells and drawing any relevant conclusions in this regard is impossible.

Contrary to the above discussed model of enforced N-cadherin expression, cells of colon cancer primocultures showing high N-cadherin expression differed in a number of ways from primoculture cells where N-cadherin expression was absent. These included different rates of proliferation and migration, varying expression profiles of select EMT markers as well as responses to chemotherapeutics where N-cadherin expression correlated with lower sensitivity of cells to model chemotherapeutics as acknowledged before [39]. When implanted to athymic mice, surprisingly, N-cadherin-positive cells produced smaller tumors than their N-cadherin-negative counterparts that, in addition, excelled in some other measured characteristics (expressions of proliferation, motility and drug resistance markers) too. Furthermore, mutation profiles revealed the presence of distinct mutations in both types of tumors and, in particular, unique spectra of miRNAs where in total 109 miRNAs were found differentially expressed with important targets mapped to crucial cellular processes including cell proliferation, DNA expression and metabolism. Thus, summarized differences indicate that high versus low N-cadherin expression in these primoculture cells occurs in the context of completed or at least operational EMT process where N-cadherin is an important but not the exclusive player. An integral element of the biological existence of such cells is their propensity to constant changes as demonstrated by developed mutation(s) upon implantation along with changed expression of some invasive markers. That this process is not straightforward and may not predict biological behavior is reflected by opposite trends in the expression of proliferation, migration and drug-resistance markers that became prominently present in tumors originating from cells with no N-cadherin expression. These seemingly conflicting findings could be at least partially reconciled in view of detected complex miRNA-related regulatory circuits in particular cells whose detailed understanding is one of the key targets for future screening and research.

## 5. Conclusions

In summary, the present results indicate that while the enforced N-cadherin expression changes some aspects of colon cancer cell phenotype, it fails to successfully drive the entire EMT process with conclusive gained prosurvival and aggressive behavior of thus manipulated cells. This conclusion can be supported by several lines of facts as evident from analyzed differences in cells as well as in tumors arising from their implantation in athymic mice. On the other hand, when N-cadherin expression changes occur in the context of spontaneously developing colon cancer, the general profiles of key characteristics in examined cells including their migratory, invasive and drug resistance features correspond to their acknowledged status, both in vitro as well as in vivo. Mutation and miRNA profiles further underlie an inherent propensity of these cells to changes that are not always straightforward and predictable.

Lastly, it is necessary to state that the robustness of the conclusions of the present study is potentially weakened by several factors such as the small number and variability of employed samples, cultivation models used as well as not used interventional analytical approaches. In particular, the status and consequences of N-cadherin overexpression were not sufficiently mechanistically researched, which could have explained observed biological differences between stabilized colon cancer cell lines and colon cancer primocultures. Despite these drawbacks, this study brings new facts concerning the functional axis of N-cadherin expression and related biological features of colon cancer cells and highlights the colon cancer primocultures as useful models for such studies.

## Figures and Tables

**Figure 1 cancers-14-05146-f001:**
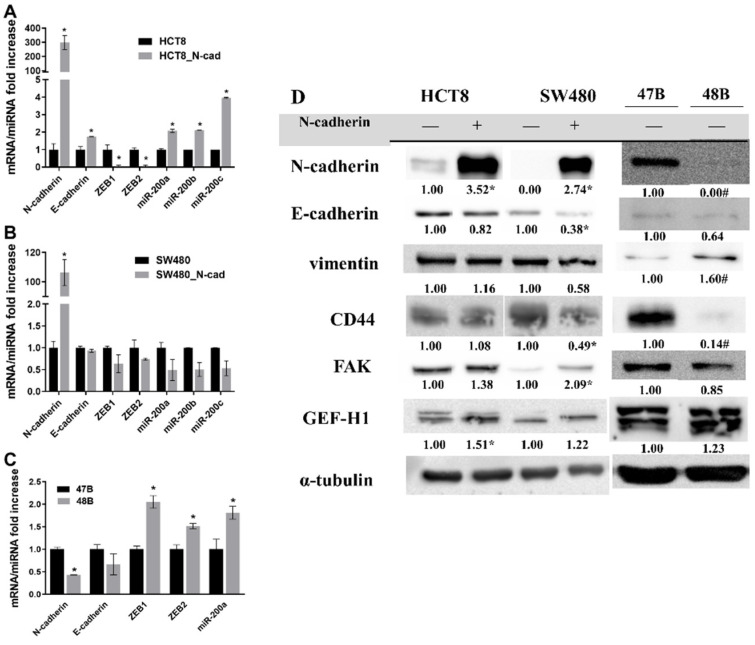
Comparison of miRNA, mRNA and protein levels of selected epithelial to mesenchymal transition (EMT) and cell migration potential markers as measured by RT-PCR and Western blot analysis (Materials and methods). (**A**) mRNA and miRNA of EMT markers in HCT8 and HCT8_N-cad cells; (**B**) mRNA and miRNA of EMT markers in SW480 and SW480_N-cad cells; (**C**) mRNA and miRNA of EMT markers in 47B and 48 primocultures; (**D**) protein expression of selected EMT and cell migration potential markers in HCT8, SW480 and its _N-cad variants, 47B and 48B primocultures. Numbers indicate densitometric analysis results. B2M was used as a reference gene in RT-PCR. * *p* < 0.05 N-cadherin plasmid-treated vs. untreated cells, resp. 47B vs. 48B cells. α-tubulin was used as a reference protein in immunoblotting. * *p* < 0.05 plasmid N-cadherin-treated vs. untreated cells, resp. 47B vs. 48B; # *p* < 0.05 47B vs. 48B. Original western blots can be found in Appendix A.

**Figure 2 cancers-14-05146-f002:**
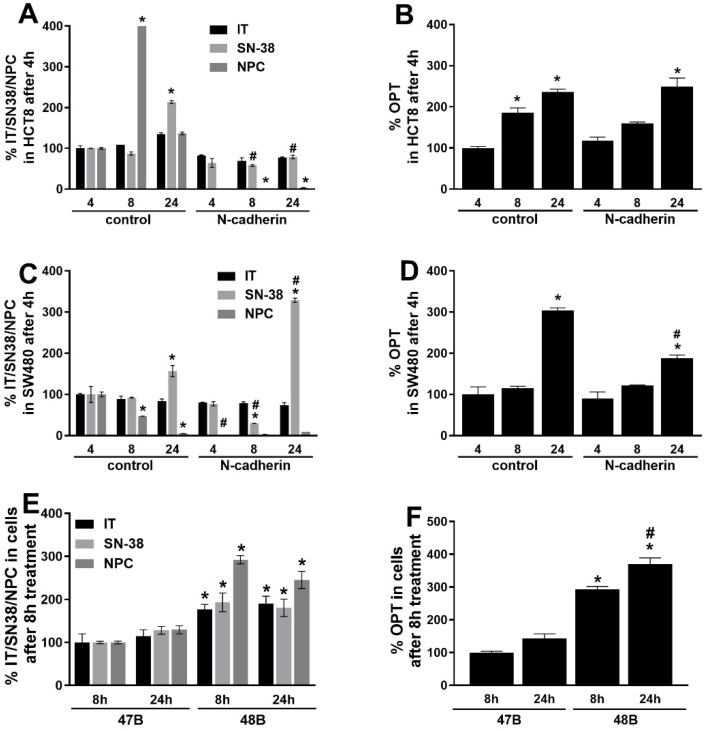
LC-MS analysis of irinotecan (IT) and its metabolites SN-38 and NPC, resp. oxaliplatin (OPT) inside the cells with different levels of N-cadherin: (**A**) amount of IT, SN-38 and NPC inside HCT8 and its N-cadherin plasmid-transfected variant; (**B**) amount of OPT inside HCT8 and its N-cadherin plasmid-transfected variant; (**C**) amount of IT, SN-38 and NPC inside SW480 and its N-cadherin plasmid-transfected variant; (**D**) amount of OPT inside SW480 and its N-cadherin plasmid-transfected variant; (**E**) amount of IT, SN-38 and NPC inside derived primocultures 47B (with naturally higher N-cadherin level) and 48B (with naturally non-N-cadherin level); (**F**) amount of OPT inside derived primocultures 47B (with naturally higher N-cadherin level) and 48B (with naturally non-N-cadherin level). The data are expressed as percentage of IT, SN-38, NPC or OPT inside the cells per mg of protein. Amount of all tested drugs was determined by LC/MS analysis. Measurements were performed in two independent experiments. * *p* < 0.05 8 h, resp. 24 h time interval vs. 4 h time interval (24 h vs. 8 h time interval in 47B and 48B cells); # *p* < 0.05 N-cadherin plasmid treated vs. untreated cells.

**Figure 3 cancers-14-05146-f003:**
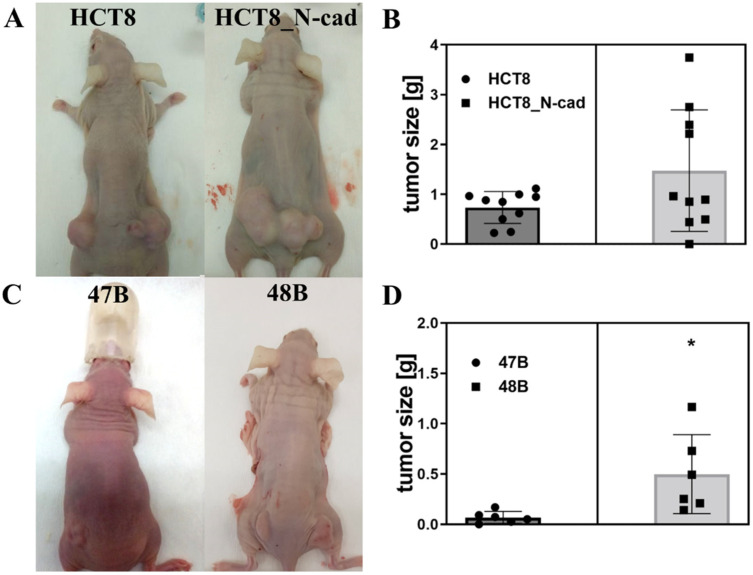
Comparison of tumor growth in Foxn1-nu mice after implantation of colon cancer cell lines and derived primocultures with different levels of N-cadherin. (**A**) mice, 21 days after implantation of HCT8, resp. its N-cadherin plasmid-treated variant; (**B**) comparison of tumor size 21 days after implantation of HCT8, resp. its N-cadherin plasmid-treated variant (n = 10/set); (**C**) mice, 40 days after implantation of derived colon cancer primoculture cells 47B and 48B; (**D**) comparison of tumor size, 40 days after implantation of derived colon cancer primoculturec cells 47B and 48B (n = 6/set). The data present means of ten tumor weight obtained from five mice per tested group in HCT8, resp. six tumor weight obtained from three mice per tested group in 47B or 48B, and their 95% confidence interval values are shown as mean ± SD. * *p* < 0.05 48B vs. 47B tumor size.

**Figure 4 cancers-14-05146-f004:**
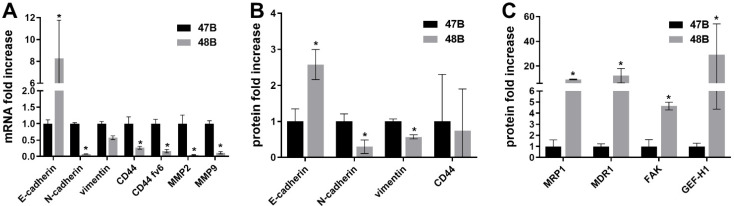
RT-PCR and Western blot analysis of tumors 47B and 48B excised from athymic immunodeficient mice. (**A**) mRNA fold increase of EMT markers in 47B and 48B tumors; (**B**) protein fold increase of markers related with EMT; (**C**) drug resistance and cell motility markers in 47B and 48B tumors; B2M was used as a reference gene. ß-actin was used as a reference protein. * *p* < 0.05 means 47B vs. 48B tumors.

**Figure 5 cancers-14-05146-f005:**
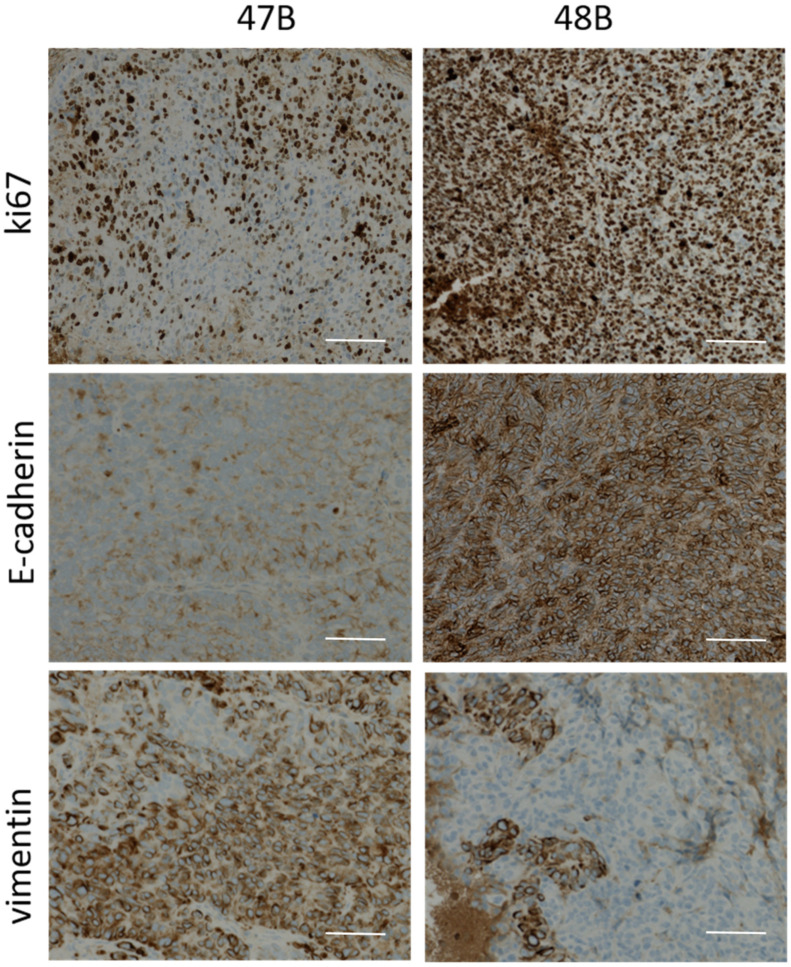
Expression of Ki-67, E-cadherin and vimentin in tumor samples 47B and 48B excised from tumor-bearing mice detected by immunohistochemistry as described in Materials and methods section. Bright-field microscopy 100×. Scale 50 µm.

**Figure 6 cancers-14-05146-f006:**
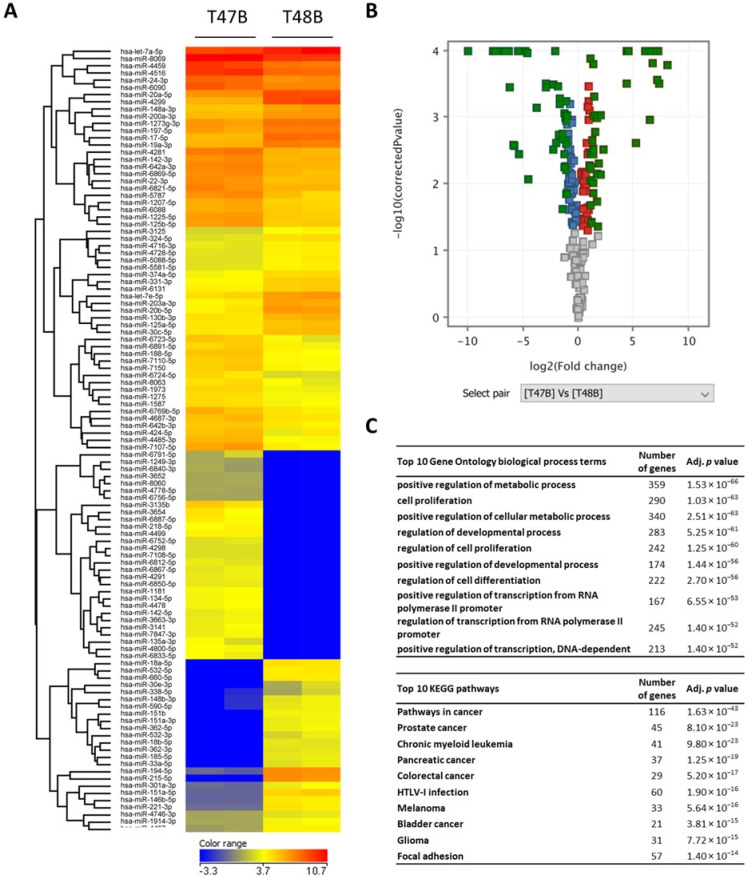
Heatmap with the unsupervised hierarchical clustering of differentially expressed miRNAs between mice-implanted cells of 47B and 48B colon cancer primocultures ((**A)**, volcano plot of miRNA array results (differentially expressed miRNAs are green colored) (**B**), and top 10 significant Gene Ontology (GO) biological process terms and KEGG pathways (**C**).

**Table 1 cancers-14-05146-t001:** IC50 values (µg/ mL) of irinotecan (IT) and oxaliplatin (OPT) treatment in original SW480 and HCT8 cell lines, their N-cadherin-transfected variant SW480_N-cad, resp. HCT8_N-cad and two colon cancer primocultures 47B and 48B isolated from patient’s samples after 48 h treatment. Data are expressed as an average ± SD.

	HCT8	HCT8_N-cad	SW480	SW480_N-cad	47B	48B
IT	3.989	1.588	32.54	7.006	259.9	13.33
OPT	5.892	0.637	1.021	0.290	216.8	24.08

**Table 2 cancers-14-05146-t002:** Molecular characteristics of tumor samples 47B and 48B excised from tumor-bearing mice. * is specific character of labeling of HGVS; * = translation termination codon.

	Gene	HGVS	Alt %
**47B**	*APC*	NM_001127510.2:c.4012C > T, p. (Gln1338 *)	99.04
	*KRAS*	NM_004985.3:c.35G > T, p. (Gly12Val)	88.95
	*TP53*	NM_001126112.2:c.818G > A, p. (Arg273His)	99.29
**48B**	*PIK3CA*	NM_006218.2:c.1633G > A, p. (Glu545Lys)	40.72
	*APC*	NM_001127510.2:c.4248del, p. (Ile1417Leufs × 2)	47.65
	*APC*	NM_001127510.2:c.6496C > T, p. (Arg2166 *)	96.21
	*RET*	NM_020975.4:c.2939T > C, p. (Met980Thr)	47.14
	*KRAS*	NM_004985.3:c.38G > A, p. (Gly13Asp)	47.3
	*TP53*	NM_001126112.2:c.1101-2A > C, intronic, splice	49.92

## Data Availability

The data presented in this study are available in this article and Appendix A.

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
