# Peer review of "Role of N-Cadherin in Epithelial-to-Mesenchymal Transition and Chemosensitivity of Colon Carcinoma Cells"

_cancers, 2022, doi:10.3390/cancers14205146_

Round 1

Reviewer 1 Report

Overall the study is well presented. The findings uncover that N-cadherin has both cell lines and primocultures differences (not unexpected, but nevertheless interesting), and demonstrate colon cancer primocultures as useful models.

Fig legends to S4 should be clearer (cells, proliferation and migration tagged to the different panels)

Page 9 in the manuscript; Fig SD must be wrong

Author Response

We would like to thank the reviewer for the comments and suggestions aimed at improving our manuscript. All of them were taken into consideration, changes were made and manuscript was revised.

1) Fig legends to S4 should be clearer (cells, proliferation and migration tagged to the different panels) 

The Fig. S4 was modified - proliferation and migration heads were provided. Names of cells are, in our opinion, not necessary to state extra since they are indicated in the graphs themselves.

2) Page 9 in the manuscript; Fig SD must be wrong

We presume this comment refers to the supplementory Fig. 4S- part D where migration of SW480 cells is shown. In fact, these cells as stated in results section did not show any migration activity whether in native (control) state or with upregulated N-cadherin expression. It might seem odd or wrong but it is just report of our results.

3) Manuscript language - spell check is required.

Manuscript was thoroughly proofread, typographical errors corrected, style and form of presentation unified. All changes made are indicated in red color.

Reviewer 2 Report

In this study, in vitro and in vivo models were used to investigate N-cadherin's role in epithelial-to-mesenchymal transition (EMT) and associated aggressive phenotypes in colon cancer cells. The experimental results highlight that the primary cell model is more suitable for the functional study of N-cadherin, and attempts are made to explain the mechanism of phenotypic differences such as drug resistance with gene mutation and miRNA profiling. This manuscript is rich in content, logical and reasonable, and is an innovative and useful research. But I have a little doubt that the N-cadherin functional rescue experiment is missing, and that there is not enough analysis of conflicting results in cell line and primary cell culture experiments.

Author Response

We would like to thank the reviewer for the comments and suggestions aimed at improving our manuscript. All of them were taken into consideration, changes were made and manuscript was revised.

1) I have a little doubt that the N-cadherin functional rescue experiment is missing, and that there is not enough analysis of conflicting results in cell line and primary cell culture experiments.

We agree with reviewer that our reported biological differences between employed stabilized cell lines and primocultures as related to differing expression levels of N-cadherin are significant. It is also true that we have not analyzed in detail nature and consequences of N-cadherin overexpression which might, perhaps, shed more light into these mentioned differences. We have not expected to find them in such an extent since target overexpression is routinely used in mechanistical studies of the same or similar kind. We understand it is relevant claim, however, it was not our primary aim in this work. We will pursue it further in our future work. In addition, we have quoted this limitation in the conclusion part of our manuscript.